

# Steady State Non-isothermal Well Flow in a Slanted Aquifer: Mathematical formulation and Field Application to a Deep Fault in the Xinzhou Geothermal Field in Guangdong, China

Guoping Lu[1*], Bill X. Hu[1*]

[1]Institute of Groundwater and Earth Sciences, Jinan University, 601 Huangpu Avenue
West, Guangzhou, Guangdong 510632, China

*Correspondence to:  guopinglu@yahoo.com; Tel.: +86-18923379841.

*Correspondence to:  bill.x.hu@gmail.com;    Tel.: +86-18624375631.





**Abstract.** This paper develops a novel mathematical formulation for geothermal well
flow. Non-isothermal flow would implicate the effectiveness of gravity as a body force
term regulated by viscosity and density as well. Consequently it is a critical concept in
practice that a dome-shaped water head surface would be present in its equilibrium-state
water potential, as a proper observation needed to understand geothermal flow fields.
Tabulation and formula are compiled on water density and viscosity as a function of
temperatures and pressures to facilitate calculations. The derived formulas were applied
to the field study site in a deep fault zone at a geothermal field in coastal Guangdong
province, China, based on observations from a thousand-meter-depth borehole drilling
project. The deep fault is unique in having a steep plane that emerges at the ground
surface, and constitutes fast flow path for deep thermal water up to $115^{o}$C and static water
pressure up to 10 MPa at the borehole bottom. The fault is conceptualized as an inclined
thin aquifer, and formula are derived for thermal outflows for the sloped aquifer to
quantify the flow in the fault plane. Results showed that the deep fault has permeability
equivalent to clean sands and lower end of unconsolidated gravels. Deep faults could
provide useful information on pathways of preferential fluid flows. The deep fault study
has several implications in deep geothermal environments and pressure characterizations,
regional groundwater circulation limits, and pressure wave propagations in earthquake
prediction in the deep crust.

Key words: deep fault, hydraulic conductivity, geothermal flow, slanted aquifer, well

flow

**1 Introduction**



This paper developed mathematical capability to characterize and subsequently
discussed its application to the deep fault zone properties in the Xinzhou geothermal field,
in coastal Guangdong province of China. The hydraulic property of the deep fault zone
were characterized with newly derived formulations for slant aquifer in this paper and
based on data from a thousand-meter-deep borehole drilling for the geothermal water
flows.
In the Xinzhou geothermal field site, the deep fault plane constitutes the fast flow path
of the thermal water. The water temperature is up to 115°C at the borehole water column,
with the static water pressure reaching up to 10 MPa at the bottom of the thousand meter
borehole. Traditional formulation using the concepts in hydraulic heads is no longer an
appropriate independent variable due to the density variant. Another consideration is with
dependence of viscosity on temperatures. The research problem faced is how to factor in
the buoyancy and viscosity to appropriately characterize the thermal well flow.
Geothermal energy comes in the forms of conductive heat and advective water flow
(Bodvarsson and Tsang, 1982). The advective geothermal flow is inherently affected by
thermal properties such as lighter density and lower viscosity at elevated temperatures
mainly in the form of buoyancy. Geothermal energy is a realistic and emerging source of
clean, renewable energy (Baioumy et al., 2015; Regenauer-Lieb et al., 2015; Craig et al.,
2013; Younger and Gluyas, 2012; Zhang et al., 2004).
Ground water flows along a fault which is a slant confined fault zone aquifer (Lu et al.,
2017; Holland, 2012; Fowles and Burley, 1994). The fault aquifer renders spatial
positioning a necessity in quantifying the well flow. Vertical  aquifers (Anderson, 2006)
is an special case of slanted aquifer. Sloping fault aquifers have additional complexity of


flow path in terms of its presentations (Huang et al., 2014; Antonio and Pacheco, 2002).
Earlier approach uses a moving boundary to approximate the lowering of the free surface
portion of the aquifer from progressive drying up the aquifer during pumping. A
mathematical model was later developed for simulating the hydraulic head distribution in
response to pumping in a sloping fault zone aquifer under a water table boundary
condition (Huang et al., 2014). Analytical solutions of seawater intrusion was developed
for sloping confined and unconfined coastal aquifers (Lu et al., 2016). And a semi-
analytical solution was presented for groundwater flow due to pumping in a leaky sloping
fault-zone aquifer surrounded by permeable matrices (Zhao et al., 2016).

Along the line of aquifer dipping, wedge-shape aquifer was found not able to be

estimated by the flow in an aquifer of uniform thickness (Hantush, 1962). Pumping in a
sloping unconfined aquifer overlaying a leaky artesian aquifer was studied as a two-
dimensional steady-state groundwater well flow (Hantush, 1964). Slant well in horizontal
unconfined aquifers was investigated for both instantaneous drainage and delayed yield
(Zhan and Zlotnik, 2002). Groundwater flow across an less permeable fault than wall
rocks was simulated using simple analytical solutions for steady-state horizontal flow
across three domains linked by requiring continuity of head and flux (Haneburg, 1995).
All these approaches deal with shallow confined aquifers, which appear limited in
thermal effect, and therefore inappropriate for geothermal water flow in deep faults.
Deep geothermal water flows are characteristic of elevated temperatures with lighter
density and lower viscosity (Xu et al., 2002; Pruess et al., 1999).

In deep faults, the geothermal water is predominantly driven by buoyancy from

elevated temperatures. Cumulated effects from lighter density would result in less



hydrostatic pressures, inducing thermal waters flowing into the fault zone or fracture
zone (Lu et al., 2017). Buoyancy inherent with fluid flows often makes the groundwater
emerge as thermal spring up onto the ground surface.
Geothermal waters more likely find their ways through deep faults to flow toward a
shallower depth. The temperature effect is pronounced during deep borehole drilling, in
which colder circulating drilling water dynamically interacts with the hot geothermal
water in surrounding wall rocks (Lu et al., 2017). We want to find out how the
temperature effect plays out in the dynamic geothermal water flow to wells of deep fault
geothermal field.
In practice the density factor is critical in understanding geothermal flows in a
geothermal field. The density factor comes in play as buoyancy force and results in a
dome-shaped feature of hydraulic heads. As a result, recognization of this arched head
surface is a must to correctly characterize the geothermal flow field, and the fluid flow
and interaction of drilling fluids with the thermal waters.
We developed analytical solution to account for the significant temperature effect on
the geothermal water flow in the sloping fault zone aquifer, and subsequently applied the
well flow equations to the field observation data of the borehole water columns to derive
the deep fault permeability data.
Our study is the first of this kind for non-isothermal flows in terms of analytical
approach for a slant confined aquifer and its applications to a deep fault. To the best of
our knowledge, no relevant analytical equations regarding thermal water flows for fault
planes in general field site applications have been found in literatures. Our paper is
focused on deep aquifers both with profound thermal effect and fault properties. The



research solutions provide tools in quantifying thermal flows and aquifer properties, and
the results provide basis and leads to researches on deep thermal and mechanical
processes as groundwater circulations and pressure water propagations in the deep crust.

The thermal waters have lower density at a higher temperature as thermal expansion

outgains compression in the crust (Pruess et al., 1999). The water density could be
slightly affected by the mineral content in salty water and the dissolved gases (Pruess et
al., 1999). The geothermal waters tend to get more acidic at an elevated temperature (Lu
et al., 2015) at a greater depth.

This paper has several objectives: 1. derivation of an analytical solution for steady

radial flow of a borehole for geothermal water in a horizontal aquifer. 2. derivation of
analytical solution for radial flow for geothermal water in a dipping aquifer. 3.
compilation of density and viscosity data for thermal waters. 4. study of the hydraulic
properties of the deep fault in the Xinzhou geothermal field site.

**2  Regional Geology and Site Description**
**2.1 Geological setting**

The Xinzhou coastal geothermal field is a part of the coastal China geothermal belt,

extending from the south to southeast and continue to east China along the coast. This
coastal thermal belt is relayed to the west to the Mediterranean-Himalayas geothermal
belt in Yunnan, Tibet, and west Sichuan of southwestern China (Wang et al, 2016; Guo
and Wang, 2012; Liao and Zhao, 1999).



In Guangdong Province, geothermal fields occur in a pattern that reflects the controlling
tectonic structures (Figure 1a). Oriented north-south and east-south deep faults are crossed by
northeastern strike faults. The Enping-Yangjiang deep fault, located on the western side of the
Pearl River estuary, has been observed cut 20 km deep into the crust (Ren et al., 2011).
The study field site is located in the southwestern coastal area in Guangdong province
(Figure 1). It is about 19 km away from the coast line and 10 km from the tidal reach of a local
river called Shouchang River (Figure 1b). The altitude of Xinzhou geothermal field is about 10 m
to 13 m (Figure 1).
Xinzhou geothermal field sees outcrops of Yanshan II granite in the southern and
sporadically in the north. The granite forms the northwestern edge of the Xinzhou granite
batholith (with an outcrop area of 292.6 km$^2$). In the Xinzhou basin, the basement rocks are
overlain by Quaternary clastic and marine sediments (Figure 1c). At the periphery of the batholith,
to the north of the site is the boundary with Precambrian-Cambrian light metamorphic clastic
rocks (Figure 1c).

**2.2   Xinzhou geothermal field site and drillings**

In the Xinzhou field site, hot springs are exposed along the stream bed of the upstream of the
coastal Shouchang river (Figure 2). Hot geothermal waters outflow from the earlier drilled
boreholes (Figure 2). Hottest 98.4$^{o}$C outflow (98.4$^{o}$C)  is found in the Jia well located in the
middle of the field and the water (Figure 1c).
A deep fault occurs at a high angle (almost vertical at 85$^{o}$) dipping to the south. The fault
was initially revealed in drillings at an earlier time (Liang, 1993), recently further characterized
by the geophysical method of Audio Magnetotelluric Sounding (AMT) (Wu, 2013). The faulting
is revealed to extend about as deep as 10 km into the crust, based on apparent resistivity and



impedance phase surveys (Wang et al., 2015), and confirmed in a 1000-m scientific drilling in
2013 (Wang et al., 2015).

The circulating water column inside a  on-going drilling borehole was generally cooler than

the borehole wall rocks, creating a higher hydrostatic pressure against the borehole wall. This
practically has become a technical utility in overcoming pressurized hot water flowing out of a
borehole. Hot water could be triggered to flow out of a borehole when a water-transmitting fault
or fracture zone is crossed in drilling. This eruption of flowing hot water comes with an
overpressure yields a hydraulic head above the ground surface, which has to be suppressed to
order to resume drilling. When down dripping starts, injection of colder drilling water gradually
cools the hot water in the borehole, creating a water column of an increasingly higher static
pressure and eventually lowering the hydraulic head level to below the ground surface. This
effectively suppresses the surging of high-temperature geothermal water. This also serves as the
working theory to resume drilling after a thermal eruption, by injecting circulation water into the
borehole when starting the down tripping. More and more circulation water into the borehole
would eventually lead to suppression of the surging of hot geothermal water.

The borehole temperatures were measured to monitor the thermal gradients for the thousand-

meter borehole (Tables 2 and 3; Figure 2). The temperature profiles were characterized before
and after the thermal eruption triggered from drilling past the fault plane.


**3  Hydraulic properties of geothermal waters**

Geothermal waters have variable density and viscosity (Table 1). Generally the

density becomes smaller at an elevated temperature (Wagner, 1999; Keenan et al., 1969;
and International Formulation Committee, 1967). It is noted that the list is specified for
saturated pressures. In comparison, also listed is the density values at pressures of 5 MPa





larger over the saturated pressure. The density becomes slightly larger at a higher
pressure within the first ten MPa (Table 1).

For a simple calculation, the density can be interpolated from the those listed in Table

1. A more rigorous approach is using numerical calculation of the density value for a pair
of given temperature and pressure. In this paper the calculation was also performed using
a numerical code modified from the module for density in Tough2 simulator (Pruess et al.,
1999). The permeability values reported in Table 2 were computed using the density
computed by the revised code. Note that the two methods above yield density values
within 0.52% difference and either one is deemed as appropriate.

Viscosity of water is lowered at a higher temperature (Table 1) (Sengers and Watson,

1986). The viscosity fitted for temperatures is good within 0.88% at saturated pressures;
and 0.25% at 5 MPa over saturated pressures for temperature between 10-200$^{\circ}$C. At 0$^{\circ}$C
the fit yields under-prediction of the viscosity with error -2.7%, and at 300-350$^{\circ}$C has an
over-prediction.

Viscosity is affected by pressure in two trends. Higher pressures lead to slightly

smaller viscosity for low temperatures (0-25$^{\circ}$C), and result in slightly larger viscosity for
higher temperatures (around >25$^{\circ}$C).

Both water density and viscosity are more subjective to temperature variations and

the pressure appears less a factor. For a 5 MPa pressure increase over the saturated
pressure, the density gains less than 0.39%, and the viscosity changes less than 0.89% for
temperatures in the range of 0-200$^{\circ}$C.





The water column will expand and rise to a higher level when being heated up. This
explains that an aquifer has a dome shape pressure surface in a geothermal field at steady
state condition (Figure 3a).


**4  Generalized Darcy's Law**
The Darcy's law can be written in the generalized dynamic form as flux in an unit
area (Brownell et al., 1977; Hubbert, 1957, 1940):

$$\mathbf{q} = - \, \mathrm{k} \, \nabla [\frac{1}{\mu} (P + \rho \mathbf{g} z)] \tag{1a}$$


$$\mathrm{k} \, \nabla [\frac{1}{\mu} (P + \rho \mathbf{g} z)] + q_s = S_s \frac{\partial P}{\partial t} \tag{1b}$$


$$\mathbf{v} = \frac{\mathbf{q}}{\theta} \tag{1c}$$


where $\mathbf{q}$ is the fluid's mass flux vector or called Darcy velocity (kg/m$^2$/sec), k is the rock
permeability (m$^2$), $\rho$ is the fluid density (kg/m$^3$) dependent of temperature and pressure
(Section 3). $\mu$ is kinematic viscosity (m$^2$/s) dependent of temperature and pressure.
Kinematic viscosity $\mu$ is related to dynamic viscosity $\nu$ (the SI unit, kg/m/s, or Pa s)
through $\mu = \nu/\rho$. $\nabla$ is partial derivative respective to coordinates, g is the gravity
constant (9.80665 m/s$^2$), and z is the height relative to a reference point. The negative
sign in front of the left side stands for the flow pointing to the opposite of the gradient. v
is the average seepage velocity or pore velocity. $\theta$ is porosity of the porous medium. $q_s$ is
the sink/source term, which is the mass flow rate injected or extracted from unit volume
of the aquifer. $S_s$ is specific storage, that is the water amount released from unit volume

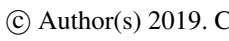



of the aquifer at one unit drop of the head. The pressure P (Pascal in kg/m/s$^2$) has the
form for constant hydrostatic pressure or variable density conditions, respectively:

$$P = \rho\, g\, (h - h_0) \qquad (2a)$$


$$P = \int_{\rho_0 h_0}^{\rho_m h_m} g\, d(\rho\, h) \qquad (2b)$$


$$P \approx \sum_{i=1}^{m} \rho_i\, g\, (h_i - h_{i\text{-}1}) \qquad (2c)$$

where h is the hydraulic head (m), and $h_0$ is the elevation (m) at a reference point.
Summation i for *k*, and *l*, *m*, *n* thereafter are the number of discrete points along the water
column.
An useful alternative form of the Darcy's law has the simplified form for groundwater
flow:

$$Q = -K\, \nabla\!\left(\frac{P}{\rho g} + z\right) \qquad (3)$$

where K is called the hydraulic conductivity (m/s), with $K = k\, g/\mu$ for conversion to
permeability. Q is the flux across unit area (m/s, or m$^3$/m$^2$/s). Note that strictly Q is
volumetric, differing from q in mass (kg/m$^2$/s), with their conversion $Q = \rho\, q$.



**5. Non-isothermal well flow in a horizontal aquifer and an inclined confined**
**aquifer**




### 5.1 Linear thermal flows in a horizontal aquifer or an inclined confined aquifer

In the geothermal water flow, both fluid density and viscosity are dependent of
temperatures. Density affects hydrostatic pressure, and viscosity reflects water's
resistance to flow. Assuming that the temperature is linearly varied along the flow
direction. In steady state, we have the Darcy's law for the flow in a unit width as:

$$\mathbf{q} = -k\,M\,\frac{d}{dl}\,[\frac{1}{\mu}(P + \rho\,\mathbf{g}\,z)] \tag{4a}$$


$$\frac{\mathbf{q}}{k\,M}\,dl = -d[\frac{1}{\mu}(P + \rho\,\mathbf{g}\,z)] \tag{4b}$$


where M is the thickness of the aquifer. $l$ is the length variable. Flow $\mathbf{q}$ takes positive value for
flow opposite to the direction of the length variable. Integration of both sides of Eq.(4b) leads to

$$\mathbf{q}\,(\frac{L_2}{k_2\,M_2} - \frac{L_1}{k_1\,M_1}) = -\int_{L_1}^{L_2} d(\frac{P}{\mu}) - \int_{L_1}^{L_2} d(\frac{\rho\,\mathbf{g}\,z}{\mu}) \tag{5a}$$


$$\mathbf{q}\,(\frac{L_2}{k_2\,M_2} - \frac{L_1}{k_1\,M_1}) = -(\frac{P}{\mu})|_{L_1}^{L_2} - (\frac{\rho\,\mathbf{g}\,z}{\mu})|_{L_1}^{L_2} \tag{5b}$$


where L is length along the flow, with subscripts 1 and 2 indicate variable at the starting point
and ending point, respectively. The summation terms are referred to Eq. (2c). The pressure term
can be calculated accurately or approximated by Eq. (2b,c) for the water column revealed in a
borehole. The second term on the right-hand side is the additional term for gravity as a body force
exerting on the system, arising from elevation difference and compounded by the thermal effect
in terms of density and viscosity. The elevation reference point is set to either one of the
calculation points, in order to yield a correct body force term owing to an elevation difference.





### 5.2 Non isothermal radial flow in a horizontal confined aquifer

Assuming in a horizontal confined aquifer, in the non-isothermal scenario, fluid density
and viscosity are variables of temperature (Figure 4c). The flow to the pumping well can
be obtained from Eq. (1a) by accounting flow area in the radial domain:

(6)

$$q_w = 2\pi kM\,r\,\frac{d}{dr}(\frac{1}{\mu}(P + \rho g h)) = 2\pi kM\,r\,[\frac{d}{dr}(\frac{P}{\mu}) + \frac{d}{dr}(\frac{\rho g h}{\mu})]$$


where $q_w$ is the flow rate (m³/s). It is assumed that P is constant over time. With the
assumption of elevation z independent of locations, the second term on the right-hand
side vanishes because no elevation difference in a horizontal aquifer.

Integrations of pressure and density, as well as viscosity, over the radial lead to

(7)

$$\int_r \frac{q_w}{2\pi kM\,r}\,dr = \int_\mu d(\frac{P}{\mu}) + \int_{h,\rho,\mu} d(\frac{1}{\mu}\rho g h)$$

Both sides can be integrated as

(8a)

$$\frac{q_w}{2\pi kM}\ln(r_2/r_1) = \frac{P}{\mu}\Big|_{R_2} - \frac{P}{\mu}\Big|_{R_1}$$


(8b)

$$\frac{q_w}{2\pi kM}\ln(r_2/r_1) = \frac{1}{\mu}\rho g h\Big|_{H_2}^{H_{T2}} - \frac{1}{\mu}\rho g h\Big|_{H_1}^{H_{T1}}$$


(8c)

$$\frac{q_w}{2\pi kM}\ln(r_2/r_1) = \sum_{i=1}^{m}\frac{1}{\mu_i}g\rho_i h_i\Big|_{R_2} - \sum_{i=1}^{n}\frac{1}{\mu_i}g\rho_i h_i\Big|_{R_1}$$






where subscript $R_1$ and $R_2$ marking the location of the boreholes, subscript H indicating
height, subscripts 1 and 2 for locations at borehole locations x2 and x1 in the aquifer, and
subscript T for the top of the borehole water column. The pressure terms are determined
by the correspondent borehole's water column, which has temperature dependent density
and viscosity.
The water potential term is reversely proportional of the kinematic viscosity
(momentum diffusivity) as the resistance of water flow. As the viscosity becomes smaller
at a higher temperature, the water potentials lead to a net loss for an elevated temperature
around the pumping well; on the other hand, it results to a net gain for an abated
temperature at the pumping well. Eq.(16a,b) is reduced to the general case of flow in a
horizontal confined aquifer as in Eq.(7) for no variations in density and viscosity


**5.3  Non isothermal flow in an inclined confined aquifer**
Considering the non-isothermal scenario, both fluid density and viscosity are variables. In
a dipped confined aquifer (Figure 4d), the flow to the pumping well can be obtained from
Eq. 1a, by accounting flux face in the radial domain:

$$q_w = 2\pi k M r \frac{d}{dr}[\frac{1}{\mu}(P + \rho \mathbf{g} z)] = 2\pi k M r \frac{d}{dr}[\frac{1}{\mu}\rho g(h+z)] \quad (9a)$$


$$E = E_0 + z = E_0 + A x \quad (9b)$$


$$r^2 = x^2 + y^2 + z^2 = (1 + A^2)x^2 + y^2 \quad (9c)$$








where the r is defined in Eq. (9c). z is the elevation of the sloped aquifer, and can be
related to x a line through the origin (Figure 4c), defined by $z = A\,x$ at the well through the
aquifer , with E being the elevation, $E_0$ as elevation at origin (0, 0), and A as the slope.


Substituting Eqs. (9b,c) into Eq. (8) leads to

$$d\left(\frac{1}{\mu}P\right) + d\left(\frac{1}{\mu}\rho\,gz\right) = \frac{Q_w}{2\pi kM}\,\frac{x\,dx + y\,dy + z\,dz}{r^2} \tag{10a}$$


$$d\left(\frac{1}{\mu}\rho\,g\,h\right) + d\left(\frac{1}{\mu}\rho\,gz\right) = \frac{Q_w}{2\pi kM}\,\frac{(x + A^2 x\;)dx + y\,dy}{(1 + A^2\;)x^2 + y^2} \tag{10b}$$


Integration over both sides have

$$\frac{1}{\mu}\rho\,g\,h + \frac{1}{\mu}\rho\,g\,Ax = \frac{Q_w}{4\pi K M}\,\ln[(1 + A^2\;)x^2 + y^2] + C \tag{10c}$$


Integrating over radius $R_1$ to $R_2$, with corresponding $x_1$ to $x_2$, leads to:

$$\frac{1}{\mu}\rho\,g\,h\Big|_{H_2}^{H_{T2}} - \frac{1}{\mu}\rho\,g\,h\Big|_{H_1}^{H_{T1}} + \frac{1}{\mu}\rho\,g\,Ax_2\Big|_{H_0}^{H_{H2}} - \frac{1}{\mu}\rho\,g\,Ax_1\Big|_{H_0}^{H_{H1}}$$

$$= \frac{q_w}{4\pi kM}\,\ln\frac{(1 + A^2\;)x_2^2 + y_2^2}{(1 + A^2\;)x_1^2 + y_1^2} \tag{11a}$$




$$\sum_{i=1}^{k} \frac{1}{\mu_i} g\rho_i h_i|_{R_2} - \sum_{i=1}^{l} \frac{1}{\mu_i} g\rho_i h_i|_{R_1} +$$

$$\sum_{i=1}^{m} \frac{1}{\mu_{x_2}} \rho_{x_2} gAx_2|_{H_0 \to H_2} - \sum_{i=1}^{n} \frac{1}{\mu_{x_1}} \rho_{x_1} gAx_1|_{H_0 \to H_1}$$

$$= \frac{q_w}{4\pi k M} \ln \frac{(1+A^2)x_2^2 + y_2^2}{(1+A^2)x_1^2 + y_1^2}$$

(11b)


where subscript H indicates height, subscripts 1 and 2 for locations at borehole locations
x2 and x1 in the fault plane, subscript 0 for the elevation reference point (which could be
set at where the borehole crossing the fault aquifer plane for simplified calculation), and
subscript T for the top of the borehole water column. The 3rd and 4th terms on the left-
hand side are for the gravity terms from reference point $H_0$ to $H_2$, $H_0$ to $H_1$, respectively.
The above formula would reduce to simplified scenario of isothermal radial flow in Eq.
(13) below.
A general equation for thermal flow in an irregular fault plane can be written as

$$\frac{1}{\mu} \rho g h|_{H_2}^{H_{T2}} - \frac{1}{\mu} \rho g h|_{H_1}^{H_{T1}} + \frac{1}{\mu} \rho g H|_{H_0}^{H_{H_2}} - \frac{1}{\mu} \rho g H|_{H_0}^{H_{H_1}}$$

$$= \frac{q_w}{4\pi k M} \ln \left( \frac{r_2^2}{r_1^2} \right)$$

(12a)


$$\sum_{i=1}^{k} \frac{1}{\mu_i} g\rho_i h_i|_{x_2} - \sum_{i=1}^{l} \frac{1}{\mu_i} g\rho_i h_i|_{x_1} +$$

$$\sum_{i=1}^{m} \frac{1}{\mu_{x_2}} \rho_{x_2} gH|_{H_0 \to H_2} - \sum_{i=1}^{n} \frac{1}{\mu_{x_1}} \rho_{x_1} gH|_{H_0 \to H_1}$$

$$= \frac{q_w}{4\pi k M} \ln \left( \frac{r_{x_2}^2}{r_{x_1}^2} \right)$$

(12b)


where the summations are calculations of the pressure exerted to the point $x_1$ or $x_2$ in the
fault plane by the water column in the borehole, index *i* for numbering temperature
measurement points, with *k* or *l* for total number of points of observed temperatures. The
r's take the distance along the path on the irregular plane.





The corresponding isothermal equation can be written for an irregular fault plane,
given that one knows the distances $r_1$ and $r_2$ (from the well) and the heights $h_1$ and $h_2$ for
two observation points.

$$( s_1 - s_2 ) + (h_2 - h_1) = \frac{Q_w}{8\pi\,K\,M} \ln\left(\frac{r_2^2}{r_1^2}\right) \tag{13}$$


And this reduces to the commonly seen Thiem equation with the h terms cancelled out
when the slope of the aquifer plane is zero (Bear, 1972).

**6. Application to hydraulic dynamics of a deep fault**

We used the deep fault in Xinzhou geothermal field to study the hydrodynamics of
the fault zone. The Xinzhou geothermal field shows a dome-shape potential surface for
the geothermal water in the early exploration in 1983 (Liang, 1993). The flow field can
be considered at equilibrium state in terms of the geothermal flows of heat and water.
The Xinzhou deep fault is an high dip-angle fault that extends thousand meters deep
into the crust (Lu et al., 2017; Wang et al., 2015). A thousand-meter borehole was able to
penetrate the fault plane (Figure 1). There had been several existent boreholes prior to the
drilling of the thousand borehole.
Well boreholes had been drilled on the hanging wall of the deep fault before the
thousand meter borehole. Ta well penetrated the fault plane at the depth of 220 m. The
deep fault outcrops at the eastern end at Maoshui Pool, in which geothermal water oozes
from the fault down under.
The deep fault system is set to have a reference point at the penetrated spot by the
thousand meter borehole (Figure 5). The x-axis points eastward parallel to the fault plane



and the y-axis is horizontally pointing to the footwall. A cross section of the aquifer is
shown in Figure 5b. The data for the calculations are listed in Table 2.
In the calculations the boreholes used were the ones that across the deep fault plane
(Table 2). The only exception was Jia well, which was not deep enough to penetrate the
fault plane.  It is close to the fault and has well fractured wall rocks as flow paths (Lu et
al., 2017). It is believed to have been well connected to the fault plane.
The water columns inside the boreholes were used to calculate the pressure head
$\sum_i \frac{1}{\mu_i} g \rho_i h_i$ at the point crossing the fault plane. Those boreholes shallower than the
thousand-meter borehole need to calculate the height interval Hi (from the fault layer
middle point at the borehole to the reference point at the thousand-meter borehole) for the
term $\sum_{i=1}^{m} \frac{1}{\mu_i} g \rho_i \, Ax \,|_{(x_2)}$. The calculations were based on density and viscosity data
corresponding to the linearly interpreted temperature data and coordinate data (Tables 1
and 2).
In the calculation of the water density, the effect of salts and dissolved gases on the
water density was assumed to be minimal and thus be neglected. This assumption is
based on the dilute nature and lack of gassy content in the thermal waters in the field site.
And this is believed to have negligible effect on the accuracy on resultant calculated
values, considering temperature being of dominant controlling factor.
The calculated permeability values for the deep fault are on the scale of 1.0e-11 m$^2$
(Table 2). And the fault plane is approximated to be homogenous with a thickness of 1 m,
based on the thousand-meter borehole drill core and historical data for the previous



drilling. In the calculation, we assume that the fault has a much larger permeability than
the fault wall rocks.


**7. Discussions and Implications**

**7.1 Borehole hydraulic dynamic causing water flow to higher ground**
Thermal effect has showed the dynamic of the geothermal flow system. In the
thousand meter borehole drilling, relatively higher pressure head was created by the
relatively colder circulating drilling fluids (Table 3). After the drilling has reached a
certain depth, the drilling operation had paused for several days for thermal recovery
prior to temperature  profile measurement.
After stoppage of the drilling for temperature profile measurements, the thermal
recovery was progressively having made the water level rising in the borehole (Table 3).
The arising borehole water level eventually reached the top the borehole and caused
eruption of the thermal flow.
A relatively lower pressure head was created in the initial stage of the thermal
recovering stage, followed by a gradually increasing head level (Table 3, Figure 3). The
pressures evolve from initial thermal recovery stage to the eventual outflowing during the
thermal recovery, demonstrating the borehole flow field reversal from colder drilling
water and hotter geothermal water.
In the processing of drilling the drilling fluid cooled down the borehole and
subsequently the wall rocks. The cooler circulated water column in the borehole created



greater pressure head than that at hotter water. The drilling water flowed outward to the
fault zone aquifer and the fault wall rocks, creating a leaky condition. This has been
interesting that the drilling water had been observed to have flowed from the lower
ground at the drilling platform flow to the Ta well (7.1 m above the ground, Table 2).
This is evident that the lubricant in drillings was found and shown up as oil sheen in other
thermal well flows.

**7.2 Extremely large permeability of the fault zone aquifer and implications**
The calculated permeability values (3.29e-11 to 1.06e-10 $m^2$) (Table 2) are equivalent
to the median ones of unconsolidated clean sand which is in the range of 1.0e-13 to 1.0e-
9 $m^2$ (Freeze and Cherry, 1979). And it is at the lower end of unconsolidated gravel
(1.0e-10 to 1.0e-7 $m^2$) (Freeze and Cherry, 1979).
The fault permeability obtained in this study is very close to but a little larger than the
value of 1.3e-12 $m^2$ derived from large scale simulation in Lu et al. (2017). The
somewhat smaller permeability from the earlier simulation approach might result from
the relatively course resolution of the discretization, in which the thermal vent was
approximated as one borehole.
We assume that the fractured walls of the fault would not alter the basic fast flow
pattern in the fault zone. This is based on the observation that the well boreholes drilled
within the hanging wall or footwall have much small flow rates than those crossing the
fault plane (Figure 1). The wells drilled into the fractured wall rocks include Xiting well,
Dun well, Old hole and East Tang wells. They have relatively small flow rates below 1.0
L/s. Above all, the fault flow is diverted into rocks of the fractured walls rather than




being draining by the latter. We could conclude that the calculated permeability numbers
from well hydraulics are served as the lower limit defining the fault's properties.
The high permeability values of the fault zone aquifer is seemingly directly related to
fast flow path of geothermal waters. This has several potential implications. Our results
for the fault permeability could also be valid to the deeper portion of the deep fault. The
geothermal reservoir in the Xinzhou field is estimated at around 3,500 m depth, which is
linked to the borehole through the fault plane (Lu et al., 2017). The high permeability of
the fault zone indicates that the deep fault zones could have deep underground
environmental conditions favoring pressure wave propagation (Yang et al., 2015; Silin et
al., 2003). And the fast flow path in the deep crust could favor the porosity wave
propagation (Rass et al., 2018; Yarushina et al., 2015).
Another potential implication involves deep groundwater circulations through deep
faults. The fast flow paths in deep faults could channel deep geothermal waters toward
shallower depths, creating a relatively lower pressure zone in the deep underground.
Deep groundwater in wall rocks is thus favored to flow toward the fault aquifer, forming
deeper circulating groundwater. This could significantly deepen the circulation limits of
regional groundwater.

**8. Conclusions**
In a geothermal field the water density factor plays an important role in
understanding the flow field. The equipotential surface presents itself as a dome-shaped
hydraulic head surface.



We have developed a series of analytical solutions for non-isothermal geothermal
steady-state water flows to wells in a confined aquifer with horizontal and dipped layer
planes. Necessary density and viscosity data were compiled for temperatures at saturated
pressures, with additional density data at higher pressures.
The analytical approach is useful in this case because it can accommodate dipped
aquifer or dipped fault plane under non-isothermal condition rather than the horizontal
aquifer for isothermal case. The temperature effect is accounted for through water density
and viscosity.
In thermal flows, gravity as a body force term has a varied effectiveness on driving
the flow because it would be regulated by viscosity and density as well. In other words,
gravity affects thermal flow differentially with variations in viscosity and density
(equation of state).
Our findings showed that the deep fault in the Xinzhou geothermal field has a large
permeability at the scale of 1.0e-11 m$^2$. This fault property corresponds to that of clean
sands and the lower end of gravels.
The primary uncertainties in relating calculations to the real world is the steady-state
nature of the formulation. An field application is involved with thickness of the fault zone,
which may vary from place to place.
Our work represents the first study with analytical solution approach for field study
of a deep fault zone. It provides a basis for further studies of deep fault property. The
results bear implications in propagation of porosity waves and regional groundwater's
deep circulations in the deep crust by geothermal waters under higher temperatures in the
crust.







**Acknowledgments**

This study was financially supported by the National Natural Science Foundation of

China (NSFC) Grant (No.41572241), and partially by The startup fund for teacher by

Jinan University. Special thanks to the help from Guangdong Provincial Hydrogeology

Brigade for access to unpublished data and assistance with field works. We are grateful to

the editors and the reviewers for the comments to improve the manuscript.

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







Tables and Figures

Table 1 Density and viscosities of water

| T(°C)[a] | Saturated Pressure (MPa) | Density[a] (kg/m³) | Dyn. Viscosity[b,c] mPa s | Kin.Visc osity m²/s × 10⁶ | Pressure (MPa) | Density[b] (kg/m³) | Dyn. Viscosity [,c] mPa s | Kin.Visc osity m²/s × 10⁶ |
|---|---|---|---|---|---|---|---|---|
| 0 | 0.1 | 999.84 | **1.793** | 1.793 | 5.1 | 1002.35 | **1.7807** | 1.7765 |
| 10 | 0.1 | 999.70 | **1.307** | 1.307 | 5.1 | 1002.04 | 1.2923 | 1.2897 |
| 20 | 0.1 | 998.21 | **1.002** | 1.004 | 5.1 | 1000.46 | 1.0008 | 1.0003 |
| 25 | 0.1 | 997.05 | **0.8905** | 0.8931 | 5.1 | 999.27 | **0.8889** | 0.8895 |
| 30 | 0.1 | 995.65 | **0.7977** | 0.8012 | 5.1 | 997.84 | 0.7992 | 0.8009 |
| 40 | 0.1 | 992.22 | **0.6532** | 0.6583 | 5.1 | 994.38 | 0.6548 | 0.6585 |
| 50 | 0.1 | 988.03 | **0.5470** | 0.5536 | 5.1 | 990.19 | **0.5477** | 0.5531 |
| 60 | 0.1 | 983.20 | **0.4665** | 0.4745 | 5.1 | 985.35 | 0.4672 | 0.4741 |
| 70 | 0.1 | 977.78 | **0.4040** | 0.4132 | 5.1 | 979.94 | 0.4045 | 0.4128 |
| 80 | 0.1 | 971.82 | **0.3544** | 0.3647 | 5.1 | 974.00 | 0.3549 | 0.3644 |
| 90 | 0.1 | 965.35 | **0.3145** | 0.3258 | 5.1 | 967.57 | 0.3150 | 0.3256 |
| 100 | 0.1 | 958.40 | **0.2818** | 0.2940 | 5.1 | 960.67 | **0.2831** | 0.2947 |
| 110 | 0.1434 | 950.98 | 0.2526 | 0.2656 | 5.1434 | 953.34 | 0.2555 | 0.2680 |
| 120 | 0.1988 | 943.08 | 0.2302 | 0.2441 | 5.1988 | 945.58 | 0.2329 | 0.2463 |
| 130 | 0.2704 | 934.8 | 0.2112 | 0.2259 | 5.2704 | 937.37 | 0.2133 | 0.2276 |
| 140 | 0.3617 | 925.9 | 0.1951 | 0.2107 | 5.3617 | 928.78 | 0.1975 | 0.2126 |
| 150 | 0.4763 | 916.7 | **0.1825** | 0.1991 | 5.4763 | 919.78 | **0.1837** | 0.1997 |
| 160 | 0.6180 | 907.1 | 0.1691 | 0.1864 | 5.6180 | 910.36 | 0.1713 | 0.1882 |
| 170 | 0.7331 | 897.3 | 0.1586 | 0.1768 | 5.7331 | 900.48 | 0.1607 | 0.1785 |
| 180 | 1.0030 | 887.0 | 0.1493 | 0.1683 | 6.0030 | 890.25 | 0.1513 | 0.1700 |
| 190 | 1.2555 | 876.3 | 0.1411 | 0.1610 | 6.2555 | 879.54 | 0.1430 | 0.1626 |
| 200 | 1.5552 | 865.0 | **0.1344** | 0.1554 | 6.5552 | 868.35 | **0.1356** | 0.1562 |
| 225 | 2.5498 | 834.0 | 0.1187 | 0.1423 | 7.5498 | 838.23 | 0.1202 | 0.1434 |
| 250 | 3.9766 | 798.6 | **0.1061** | 0.1329 | 8.9766 | 804.47 | **0.1075** | 0.1336 |
| 275 | 5.9465 | 758.6 | 0.0976 | 0.1287 | 10.9465 | 764.54 | 0.0988 | 0.1292 |
| 300 | 8.5885 | 712.5 | **0.08592** | 0.1206 | 13.5885 | 722.39 | **0.08782** | 0.1216 |
| 325 | 12.0509 | 654.9 | 0.07600 | 0.1160 | 17.0509 | 671.01 | 0.07913 | 0.1179 |
| 350 | 16.5305 | 572.8 | **0.06609** | 0.1154 | 21.5305 | 607.59 | **0.07045** | 0.1159 |

Note: a. At saturated pressure (Wagner, 1999), critical point at 647.096K, 22.064
MPa density 322 kg/m³. b. Viscosity in bold is from Sengers and Watson (1986)
and Lide (2002), c. Dynamic viscosity calculated for (Pa s or kg/m/s) $v = A \times$
$10^{B/(D \times T - C)}$ for the temperature range (0-250°C), where $T$ is temperature in
Kelvin, for saturated pressure A = 2.4×10⁻⁵ Pa·s, B = 246 K, and C = 140 K,
D=0.995; for pressure 5 MPa above the saturated pressure $A$ = 2.4×10⁻⁵ Pa·s, B
= 257.62 K, C= 140 K, and D = 1.02.








Table 2 Well Borehole data and calculated permeability values for the deep fault at
Xinzhou geothermal field[a,b,c]

| No. | Well borehole | x(m) | y (m) | Borehole Ground Elevation (m) | Head above ground[b] (m) | T[d] (°C) | Outflow (10³ kg/d) |
|---|---|---|---|---|---|---|---|
| 1 | Maoshui Pool | 316.2 | 67 | 7.75 | 1.0 | 66.5/66.5(0) | 120 |
| 2 | Dongwei Well | 324.5 | 64.75 | 9.0 | -0.25 | 71.0/88(23) | 0 |
| 3 | Jia Well | 10.0 | 44.0 | 7.9 | 4.2 | 98/98(22) | 550 |
| 4 | Ta Well | -8.0 | 53 | 8.15 | 7.1 | 96/101(160) | 350 |
| 5 | 1000 m well borehole[f] | 0.0 | 0.0 | 7.8 | 0 | 95/107(740) | 850 |
| 6 | F1 Fault[g] | - | 67 | | | | |


| No. | Well borehole | P[e] (MPa) | $\sum \rho \ g h/\mu$[f] | k (m²)[h] |
|---|---|---|---|---|
| 1 | Maoshui Pool | 7.081 | 2.66E+07 | 1.06e-10 |
| 2 | Dongwei Well | 7.067 | 2.96E+07 | 3.29e-11 |
| 3 | Jia Well | 7.059 | 3.16E+07 | 2.15e-11 |
| 4 | Ta Well | 7.097 | 3.17E+07 | 2.16e-11 |
| 5 | 1000 m well borehole[g] | 7.055 | 2.52E+07 | - |
| 6 | F1 Fault[h] | | | 1.3e-12[i] |

Note: a. Basic data records refer to site report (Wang et al., 2015; Liang, 1993). b. Pressure calculated
for what above the 740 m depth of the 1000 m well. c. The deep fault dips southward at an angle
of 85°. d. Temperature at the top/bottom (depth at the fault plane in parenthesis) of the borehole
water column. e. Hydrostatic water pressure of each borehole relative to the point of 1000-m-
borehole intercepting the fault plane. f. Hydrostatic pressure term with viscosity effect for the
water column above reference point of the 740 m depth. g. Borehole drilling started Oct.1, 2013.
Burst Outflow 850 m³/day at 19:30 on Nov. 7, 2013, borehole diameter 0.15 m. Temperature
measurements shown in Figure 2. h. Calculations using Eq. (11b), with approximation the fault
zone plane as 1 m. i. Source from Lu et al. (2017).





Table 3. Temperature measurements for the 1000 m depth borehole[a,b,c]

| Well borehole | Head above ground (m) | T[d] ($^o$C) | P[c,e] (MPa) | Outflow ($10^3$ kg/d) |
|---|---|---|---|---|
| A1(685m) 10/30 | -2.2 | 37.0~99.2 | 6.580 | 993.4~961.7 |
| A2 (685m) 10/31 | -1.5 | 42.8~102.0 | 6.561 | 991.9~959.8 |
| A3(685m) 11/01 | -0.45 | 45.0~104.5 | 6.550 | 990.3~957.9 |
| A4(685m) 11/02 | -0.35 | 51.9~106.8 | 6.538 | 987.2~956.2 |
| B(740m) 11/09 (outflow) | +5.5 | 97.9~109.8 | 6.996 | 959.6~954.2 |
| C(1002m) 12/07 | -1.5 | 35.1~96.7 | 7.087 | 994.1~963.8 |
| C f(1002m) 12/07 | +5.5 | 98.0~113.0 | 7.055 | 959.6~954.8 |

Note: a. Basic data records refer to the site report (Wang et al., 2015; Liang,
1993). b. Pressure calculated for water above 740 m depth;  The deep
fault dips southward at angle of 85$^o$. c. The drilling fluid at temperature
around 45$^o$C, having pressure of 7.180 MPa at the reference point at
which borehole crossing the fault plane. d. Temperatures at the top and
bottom of the borehole water column. e. Borehole water pressure at the
point (depth 740 m) intercepting the fault plane. Temperature
measurements shown in Figure 2.



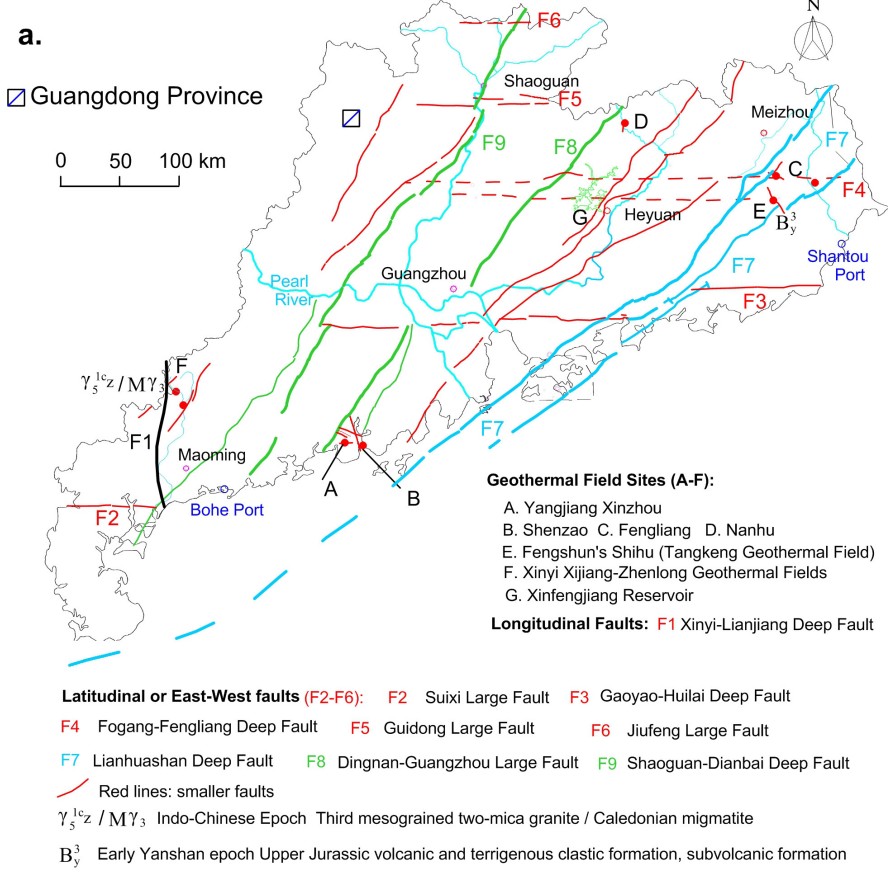






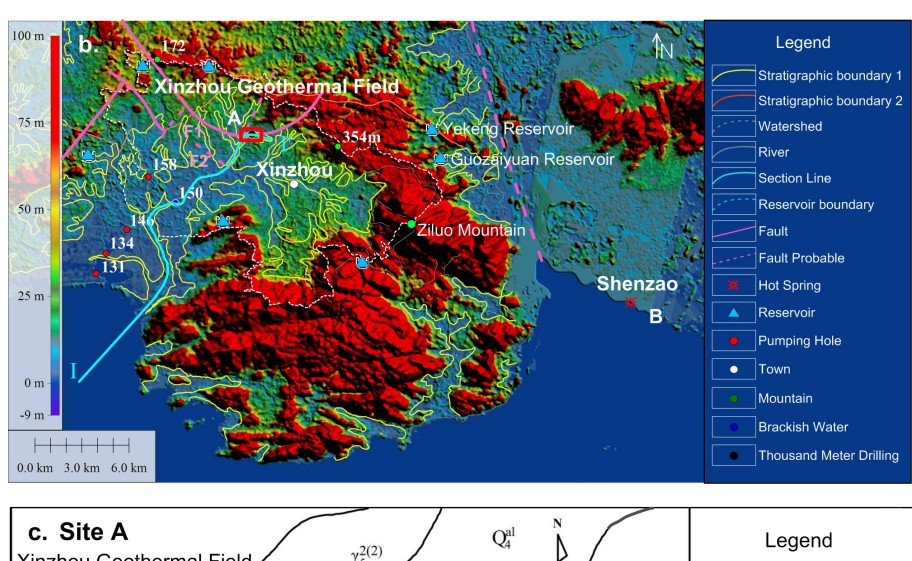

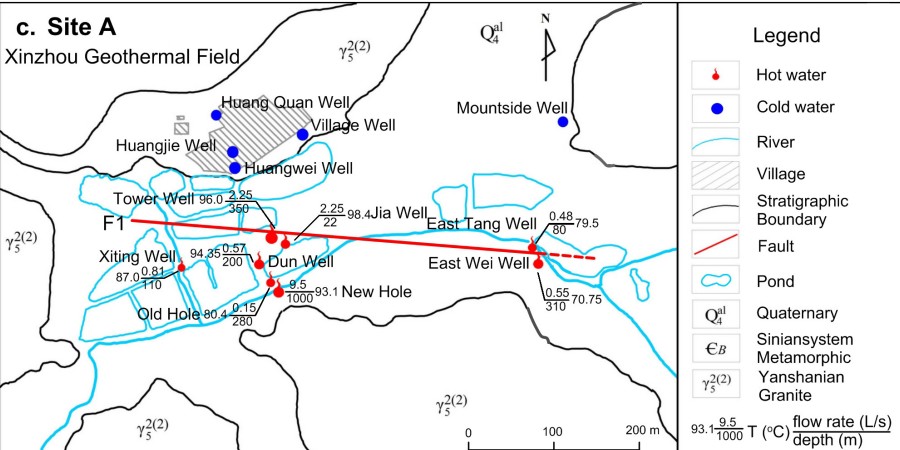

Figure 1. Geological background map for Xinzhou geothermal field in Yangjiang, Guangdong province: a. Regional tectonic map (Guangdong Province Geological Bureau Regional Geological Survey Brigade 1988; Chinese Academy of Sciences,1959); b. Regional geological map for Xinzhou geothermal field (Geological information drawn from 1: 250,000 outline configuration diagram of Yangjiang city, 2004); and c. Water sampling sites. New Hole was a 1000 m deep scientific borehole (Wang et al., 2015). Cross section I-I' shown the local stream discharge.




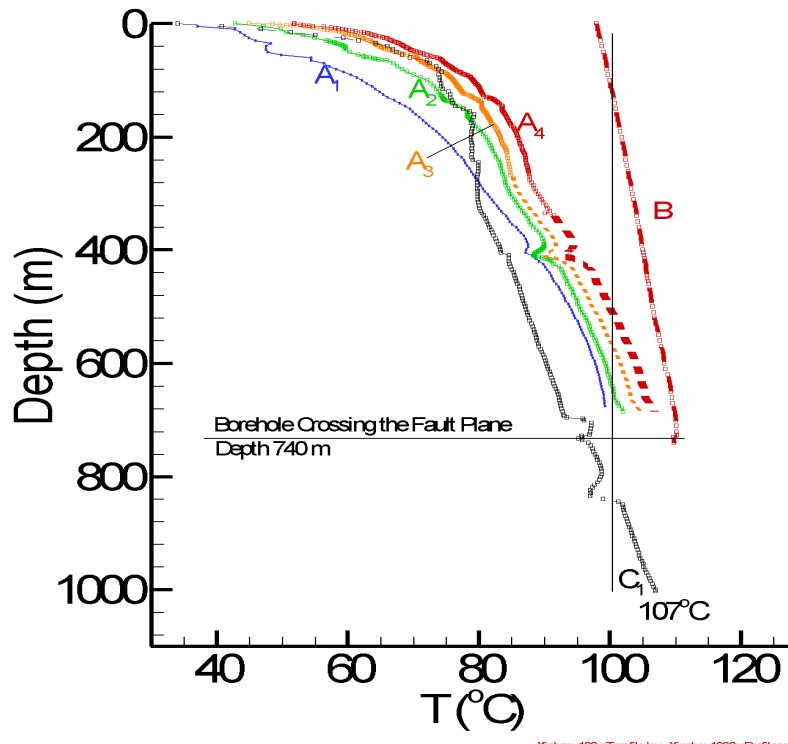

Figure 2. Temperature profiles during drilling of the 1000 m borehole (Figure 1c). A: curves
for temperature recovery at the 685 m drilling depth, with A1, A2, A3 measurements from Oct.
30, 31 to Nov.1, 2, 2013. A4: Curve prior to thermal water eruption at 740 m depth, on Nov.7,
2013. B: Curve at the thermal water eruption at 740 m depth, on Nov.7, 2013, with 109.8$^{\circ}$C
recorded at 740 m. C: curve measured on Dec. 7, 2013 for final drilling depth 1002.25 m.
Relevant statistic data in Tables 2 and 3






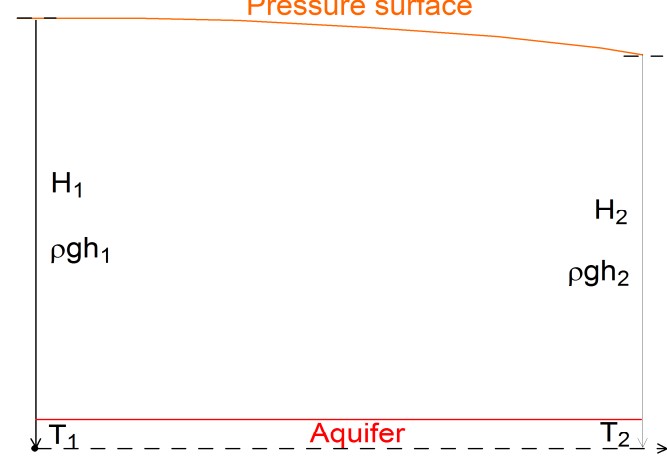

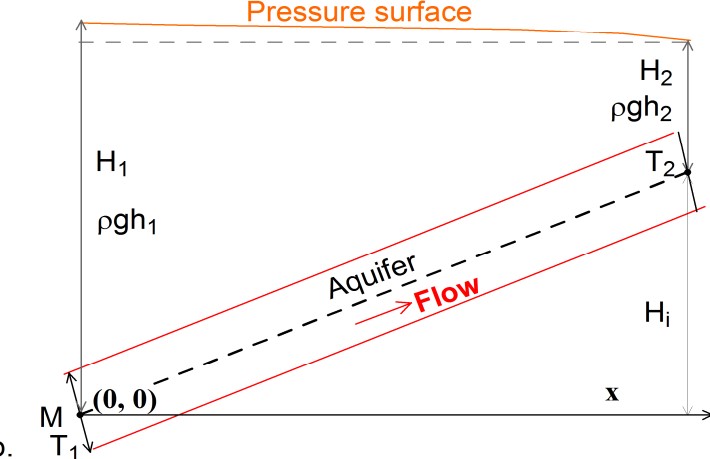

Figure 3. Geothermal water flow in confined aquifer under non-isothermal conditions. a.
Horizontal aquifer under no flow condition; b. Inclined aquifer. The origin of the coordinates
goes through the center point of the aquifer.






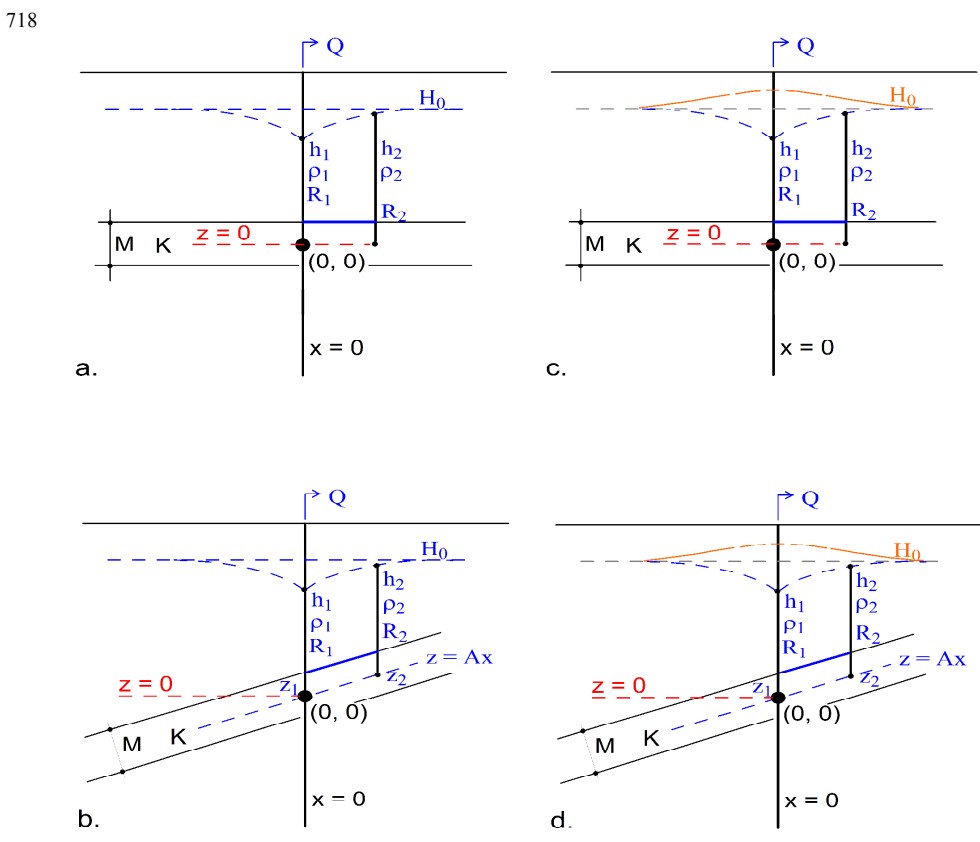


slantconfinedredo2

Figure 4. Illustration of an inclined confined aquifer under isothermal and non-isothermal
conditions. a. Isothermal horizontal aquifer; b. Isothermal inclined aquifer; c. Non isothermal
horizontal aquifer, and d. Non isothermal inclined aquifer. The origin of the coordinates goes
through the well at the center point of the aquifer. where $z = Ax$ is the median line through
reference origin $(0, 0)$ with $A$ as the slope.






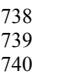

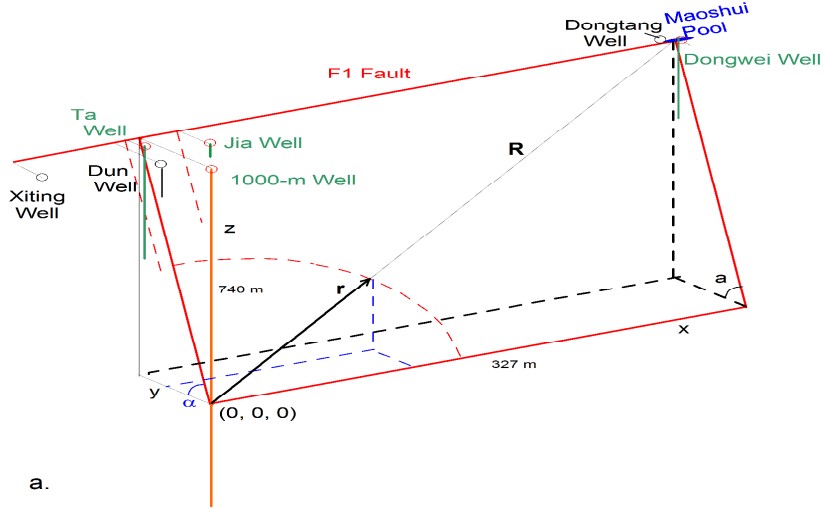

a.

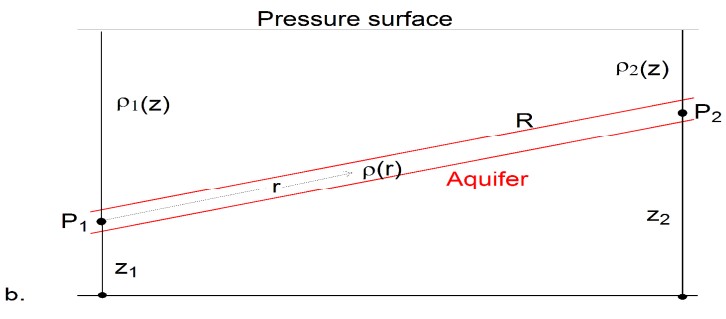

b.


Figure 5. Illustration of deep fault F1 in Xinzhou geothermal field. a. The F1 fault plane; b. Well
borehole system. $\alpha$ is the dip angle of the deep fault F1 at 85°, y-axis points to the foot wall
along the deepest gradient, and x axis the horizontal along the strike of the fault plane.
Location of the fault is referred to Figure 1c.