# Peer review of "Steady State Non-isothermal Well Flow in a Slanted Aquifer: Mathematical formulation and Field Application to a Deep Fault in the Xinzhou Geothermal Field in Guangdong, China"

_Hydrology and Earth System Sciences, 2018_

## Referee Comment (RC1) · Anonymous Referee #1 · 10 Apr 2019

In this paper an analytical solution is developed for well flow in a non-isothermal slanted aquifer. The temperature distribution is considered to be given. Thus Darcy's law and continuity equations are solved. The analytical solution is applied to the Fault in the Xinzhou Geothermal Field in Guangdong (China). The topic is interesting and fits well with the journal. My concern is with the organization and focus of the paper. I have also a technical concern regarding the developed solution. The paper is not well written. It misses structure and organization. Hence, I strongly suggest to do a thorough re-organization and possible re-writing of parts of the manuscript. Technically, I have a

problem with the way in which Darcy's law is presented. I am not sure that the viscosity should be within the gradient operator. I am not familiar with this form and I am not sure that this is correct. This may help in getting the analytical solution (especially, for the non-isothermal radial flow in a horizontal confined aquifer) as viscosity will be in the right hand side but I think this is not correct. I suggest major revisions. My comments are below: - Please justify why the viscosity is included in the gradient operator. This is essential because form me this is not correct. - Comparison against numerical solution (using an existing model) will confirm the accuracy of the developed solution - Please revise the introduction and state clearly the objective of the paper (which is for me analytical solution for non-isothermal flow). The introduction is too long. - Figures 4a and 4b not cited in the text. - Please revise: "Consequently it is a critical concept in practice that a dome-shaped water head surface would be present in its equilibrium-state water potential, as a proper observation needed to understand geothermal flow fields. " - In section 2.1 please include a vertical cross section representation of the site. - Please split Figure 1 in 2 figures - Tables 2 and 3 are cited in the text before table 1. - There is no need for table 1 which is standard in thermodynamics. - At line 192: "also listed is the density values" should be "also listed are the density values" - I did not understand why properties of saturated water are presented (+ properties at P=5.1MPa). At line 195, it is mentioned that density is function of P and T. The discussion here is ambiguous. I think variation of density and viscosity in terms of pressure and temperature is standard in thermodynamics and there are several existing models to do that (as n Tough2 simulator used in this paper). Authors can simply mention that they used Tough2 simulator and there is no need for further discussion. (Section 3 can be removed). - Please remove "The negative sign in front of the left side stands for the flow pointing to the opposite of the gradient" as it is standard. - Line 244 is not clear - Line 281 "Non isothermal radial flow in a horizontal confined aquifer 281 Assuming in a horizontal confined aquifer, in the non-isothermal scenario, fluid density 282 and viscosity are variables of temperature (Figure 4c)" Please revise this sentence and explain how the temperature is variable. - Grammatical and editing check (figure

numbering and citation, equations). - Results are not well discussed and illustrated.

---

## Referee Comment (RC2) · Anonymous Referee #2 · 15 Apr 2019

In the manuscript, an analytical solution is proposed to analyze steady state non-isothermal well flow in a slanted aquifer. The analytical solution is then applied to a case study to explain some data. The topic is interesting and is appropriate for a possible publication to HESS. However, due to several weaknesses about the organization of the manuscript, its writing, the English language or about the mathematical developments or the application to the case study, I do not recommend to accept the manuscript. It needs to be reorganized and rewritten to clarify different points before that a possible publication could be considered. Details of my reviews are given below. 1°) The manuscript is messy, difficult to read and to understand. It needs to be reorganized and much better written. For instance, in the introduction the first sentences are written like an abstract, before describing the site and the objectives of the study. But the objectives of the work appear without a clear idea about the context and about the state of the art. It seems more based on some data gathered from a field site than a clear and comprehensive understanding of a scientific question. An introduction clearly written should clearly state the context of the work, the scientific questions addressed and the method proposed to solve them. This is an example from the introduction, but the other parts of the manuscript requires also to be reorganized and clarified (see below). 2°) Furthermore, the English language needs to be thoroughly proof-read and re-written. I recommend finding a native English speaker to assist with this component. 3°) Some references are missing, especially in the field of geothermal flow associated to fault zones. Some very general references about geothermal applications are given in the manuscript, but, for instance, the following references could be more appropriately cited : Malkovsky, V. .I., and F. Magri (2016), Thermal convection of temperature-dependent viscous fluids within three-dimensional faulted geothermal systems: Estimation from linear and numerical analyses, Water Resour. Res.,52, 2855–2867, doi:10.1002/2015WR018001. Zhao, C., B. E. Hobbs, A. Ord, S. Peng, H. B. M€, L. Liu (2004), Theoretical investigation of convective instability in inclined and fluid-saturated three-dimensional fault zones, Tectonophysics.,387(1–4), 47–64, doi:10.1016/j.tecto.2004.06.007 These are only examples. About flow in slanted aquifer, some other references could be also of interest. 4°) The calculations in section 4 and 5 are simple, but the boundaries conditions used should be clarified. The aim of the manuscript is to propose a coupling between head and temperature variations, but it is not clear how varies temperature in figure 4 and 5. Line 262, it is assumed that the temperature is linearly varying along the flow direction. But why this is necessary, how this is justified ? It should be clarified and explained. Some equations should be checked, in particular equation 1b which is incorrect in this form. 5°) It is difficult to understand from section 6 and figure 5 how analytical results from section

5 are applied to the field case. This should be clarified as well as the geometry of the aquifer and the assumptions used for the calculations. 6°) How is calculated the fault permeability ? Permeability values are given (line 392) and largely discussed (section 7.2). It appears as a major result (line 45-46 in the abstract and 475-477 in the conclusion), but it should be explained how permeability is calculated. 7°) Moreover, in such permeable fault zone, with such temperature gradients, one may expect natural free convection to occur. This should be discussed since conditions for flow and pressure distribution in the fault zone (slanted aquifer) should control the response observed in the deep borehole.

---

## Author Comment (AC1) · 16 May 2019

Hu Anonymous Referee #1

Comment: "In this paper an analytical solution is developed for well flow in a non-isothermal slanted aquifer. The temperature distribution is considered to be given.

Thus Darcy's law and continuity equations are solved. The analytical solution is applied to the Fault in the Xinzhou Geothermal Field in Guangdong (China). The topic is interesting and fits well with the journal. "

Response: Thanks for the positive feedback.

Comment: "My concern is with the organization and focus of the paper. I have also a technical concern regarding the developed solution. The paper is not well written. It misses structure and organization. Hence, I strongly suggest to do a thorough reorganization and possible re-writing of parts of the manuscript. "

Response: We have done some reorganization of the manuscript in creating a new section Case study. All the relevant materials regarding the study site are moved to this case study. And we add a section in the Discussion for drilling fluid effect and implication.

We went through a thorough editing of the text to improve the flow of English.

Comment: "Technically, I have a problem with the way in which Darcy's law is presented. I am not sure that the viscosity should be within the gradient operator. I am not familiar with this form and I am not sure that this is correct. This may help in getting the analytical solution (especially, for the non-isothermal radial flow in a horizontal confined aquifer) as viscosity will be in the right hand side but I think this is not correct. I suggest major revisions. "

Response: We have re-organize part of the manuscript. Specifically, we move the text regarding site description to a newly created section Case Study. And we did a thorough editing to improve the English flow.

The viscosity and density both are state variable. Therefore it is right for them appear within the gradient operator.

The viscosity appears within the gradient operator because it is a variable dependent of temperature and pressure, which are functions of space (and time). There is no

reasonable argument to put the viscosity outside the operand.

For a verification of the analytical solution, we have tried our best to find a ready example in literature but failed to find one. The best source is the code T2Well (Pan et al., 2011) which does not have a case close to our aquifer even in the simplest horizontal scenario. To our surprise, we could not find any lead for one example, even after consulting several experts in this field (Tianfu Xu, the author of Toughreact code, Lawrence Berkeley National Lab; Keni Zhang, author of the parallel version of Tough2 code, among others) .

To do a numerical validation is not a easy task; in fact, it could take tremendous more effort than expected. This is because the coupling of thermal with flow is inherently complicated. In Tough2 code (Pruess et al., 1999), it takes an extraordinary large gridblock (e.g., 10e30) to keep the temperature constant. This in turn leads to constant pressure for the gridblock. Therefore, the approach for a realistic case needs a thermal field created for this purpose, rendering a very complicated procedure and much effort. In the regard, a validation of the numerical solution is associated with too much work.

We believe our analytical approach is sound.

We opt to not provide a numerical validation in this revision. However, we will bear this in mind and do our best to work this problem for another paper in this relevant area.

Thanks for reasonable doubt. But we beg to disagree.

Reference: Pan, Lehua, Curtis M. Oldenburg, Yu-Shu Wu and Karsten Pruess. 2011. T2Well/ECO2N Version 1.0: Multiphase and Non-Isothermal Model for Coupled Wellbore-Reservoir Flow of Carbon Dioxide and Variable Salinity Water. Lawrence Berkeley National Laboratory. LBNL-4291E.

Comment: My comments are below: - Please justify why the viscosity is included in the gradient operator. This is essential because form me this is not correct. - Comparison against numerical solution (using an existing model) will confirm the accuracy of the

developed solution

Response: Just like density, viscosity is also a state variable. If density is needed inside the gradient operator, so is viscosity.

The viscosity reflects the fact that fluid property is affected by state variables temperature and pressure. Therefore the viscosity is a dependent variable of temperature and pressure. In simplified calculations, the viscosity is expressed as constitutive relation for temperature and pressure.

The viscosity is included in the gradient operator for easing of operation. If it were taken out of the gradient, the viscosity is hard to be determined for a specific value proper in a calculation. More is that the viscosity is a measure for resistance to flowing against a "force". In this sense, it acts like permeability (regulated by viscosity) for resistance to fluid flow. Because it is a variable just like density, so they should be both placed inside the gradient operator.

Reference: Pan, Lehua, Curtis M. Oldenburg, Yu-Shu Wu and Karsten Pruess. 2011. T2Well/ECO2N Version 1.0: Multiphase and Non-Isothermal Model for Coupled Wellbore-Reservoir Flow of Carbon Dioxide and Variable Salinity Water. Lawrence Berkeley National Laboratory. LBNL-4291E.

Comment: - Please revise the introduction and state clearly the objective of the paper (which is for me analytical solution for non-isothermal flow). The introduction is too long. - Figures 4a and 4b not cited in the text. - Please revise: "Consequently it is a critical concept in practice that a dome-shaped water head surface would be present in its equilibrium state water potential, as a proper observation needed to understand geothermal flow fields. " - In section 2.1 please include a vertical cross section representation of the site. - Please split Figure 1 in 2 figures - Tables 2 and 3 are cited in the text before table 1.

Response: We state our main objective in the text..

[Figure]

We moved part of the text for field site description to the newly created field site section, and removed parts of the text.

Added citation for Figures 4a and 4b. Thanks.

Revised the text " Consequently it is a critical concept in practice that a dome-shaped water head surface would be present in its equilibrium state water potential, as a proper observation needed to understand geothermal flow fields. ". The revision is " In practice it is a critical concept that ground water hydraulic-head surface could rise to a higher level at an elevated temperature and that a dome-shaped water head surface could be present in its equilibrium-state. Consequently an elevated-water column concept is needed for the on-site observations to be properly organized to understand geothermal flow fields. "

In previous Section 2. We added a cross section for the site and split Figure 1 into 2 figures. We re-arranged the text so Table 1 is cited before Tables 2 and 3.

Comment: - There is no need for table 1 which is standard in thermodynamics. Response: It appears overdone with the data in Table 1 for the density and viscosity data with temperatures and pressures. It takes non trivial effort to put together these data as they come from several none-common sources and often with incomplete data set.

But to the contrary, the table is handy and useful in several ways: 1. It has new contribution of the authors. It provides new formula for viscosity values for saturated pressures and increased pressures, respectively. 2. It provides the trend of changes in density and viscosity with pressures, which are not changed significantly as pressure increases. 3. In comparison with the revised code for calculation of density and viscosity, the interpolated data come as handy and accurate enough.

The table and the section have been heavily revised to enhance clarity and readability.

Comment: "- At line 192: "also listed is the density values" should be "also listed are the density values" - I did not understand why properties of saturated water are presented

(+ properties at P=5.1MPa). At line 195, it is mentioned that density is function of P and T. The discussion here is ambiguous. I think variation of density and viscosity in terms of pressure and temperature is standard in thermodynamics and there are several existing models to do that (as n Tough2 simulator used in this paper). Authors can simply mention that they used Tough2 simulator and there is no need for further discussion. (Section 3 can be removed). "

Response: The editorials have been made.

The density and viscosity of the table is for completeness in calculations. And two standalone formula have been provided for viscosity calculations.

Even though it is a standard that density and viscosity are varied in terms of temperature and pressure, it is convenient and useful to provide a table for accessibility and readability of the text. The relevant data for calculations in existing models are hard to access because there is no module to direct input to obtain the desire results. For example, the viscosity data are hard-wire in the coding for TOUGH2 code.

Comment: "- Please remove "The negative sign in front of the left side stands for the flow pointing to the opposite of the gradient" as it is standard. - Line 244 is not clear - Line 281 "Non isothermal radial flow in a horizontal confined aquifer 281 Assuming in a horizontal confined aquifer, in the non-isothermal scenario, fluid density 282 and viscosity are variables of temperature (Figure 4c)" Please revise this sentence and explain how the temperature is variable. - Grammatical and editing check (figure numbering and citation, equations). - Results are not well discussed and illustrated.

Response: We made the editorials. Thanks. We revised the text as: " For the non-isothermal scenario in a horizontal confined aquifer (Figure 4c), fluid has a lighter density and a lower viscosity at a higher temperature. The resultant hydraulic head surface is thus affected. Typically a dome-shape hydraulic head surface is formed, as the aquifer contacted with heat source possesses. Lighter pore water could would rise up in a borehole to a higher hydraulic head than denser pore water. Conversely, colder

water injection into a heated aquifer could result in a funnel-shape head surface. "

We revised the case study section to provide more explanations for the results.
* * *

---

## Author Comment (AC2) · 16 May 2019

Hydrol. Earth Syst. Sci. Discuss.,https://doi.org/10.5194/hess-2018-624-RC2, 2019© Author(s) 2019. This work is distributed underthe Creative Commons Attribution 4.0 License. Interactive comment on"SteadyStateNon-isothermal Well Flow in a Slanted Aquifer:Mathematical formulation and Field Application toa Deep Fault in the Xinzhou Geothermal Field inGuangdong, China"byGuoping Lu and Bill X. Hu Anonymous Referee #2

CommentïijŽ In the manuscript, an analytical solution is proposed to analyze steady state non-isothermal well flow in a slanted aquifer. The analytical solution is then applied toa case study to explain some data. The topic is interesting and is appropriate for apossible publication to HESS. However, due to several weaknesses about the organi-zation of the manuscript, its writing, the English language or about the mathe-maticaldevelopments or the application to the case study, I do not recommend to accept themanuscript. It needs to be reorganized and rewritten to clarify different points beforethat a possible publication could be considered. Details of my reviews are given below. ResponseïijŽ We did a reorganization and a thorough editing for the manuscript. We made every effort to addressing the comments to improve the manuscript. Specifi-cally, we added assumptions for the equations. We clarified the procedures for obtain-ing the permeability data for the deep fault. We also addressed the possibility of natural convection of geothermal water in the field site of deep fall.

In summary, we made the following changes. 1. we have re-organized the text in putting the site description to a new section Case Study. 2. We added a cross section of the field site as in Figure 5b. 3. We have modified Table 1 (density and viscosity data) to make it stand out for our novel formula of viscosity at the very start of the note. 4. We added a discussion section for the effect and implication from borehole drilling fluid. The text is from the previous site description section.5. We addressed the concern that natural free convection in a geothermal field of a deep fault. And 6. We added assumptions to the analytical solutions.

We believe we have fully addressed the review comments. The paper is sound in terms of science and writing.

CommentïijŽ 1) The manuscript is messy, difficult to read and to understand. It needs to bereorganized and much better written. For instance, in the introduction the first sen-tences are written like an abstract, before describing the site and the objectives of thestudy. But the objectives of the work appear without a clear idea about the contex-tand about the state of the art. It seems more based on some data gathered from afield

site than a clear and comprehensive understanding of a scientific question. Anintroduction clearly written should clearly state the context of the work, the scientificquestions addressed and the method proposed to solve them. This is an examplefrom the introduction, but the other parts of the manuscript requires also to be reor-ganized and clarified (see below). ResponseïijŽ Thanks for your time for the comments. but the word messy is overly harsh. We have partially re-organized the text. The introduction has been re-organized and the site description has been moved to the case study to improve the structure of the paper. We revised to improve the context.We reworked the scientific problem as: "Field studies have shown that a geothermal field is active in geothermal water outflow with a central area of high temperature and a periphery of dramatically lowered down water temperatures. We want to find out how the temperature effect plays out in the dynamic geothermal water flow to wells in the deep fault geothermal field, and what method can be used to characterize the geothermal flow in the fault zone to improve understanding of the deep fault."

CommentïijŽ 2) Furthermore, the English language needs to bethoroughly proof-read and re-written. I recommend finding a native English speaker toassist with this component. ResponseïijŽ We have edited the text thoroughly to improve the flow of English. We believe the revised manuscript is sound. Thanks.

CommentïijŽ 3) Some references are missing, especially in the fieldof geothermal flow associated to fault zones. Some very general references aboutgeothermal applications are given in the manuscript, but, for instance, the followingreferences could be more appropriately cited :Malkovsky, V. .I., and F. Magri (2016),Thermal convection of temperature-dependent viscous fluids within three-dimensionalfaulted geothermal systems: Estimation from linear and numerical analyses, WaterResour. Res.,52, 2855–2867, doi:10.1002/2015WR018001. Zhao, C., B. E. Hobbs, A.Ord, S. Peng, H. B. MC, L. Liu (2004), Theoretical investigation of convective instabilityin inclined and fluid-saturated three-dimensional fault zones, Tectonophysics.,387(1–4), 47–64, doi:10.1016/j.tecto.2004.06.007 These are only examples. About flow inslanted

aquifer, some other references could be also of interest. ResponseïijŽ ïijŭe added these papers (Malkovsky and Magri, 2016; Zhao et al., 2004), to go with the ones we already have had for fault-zone flow characterizations. Relevant discussion is made. Thanks!

CommentïijŽ 4) The calculationsin section 4 and 5 are simple, but the boundaries conditions used should be clarified.The aim of the manuscript is to propose a coupling between head and temperaturevariations, but it is not clear how varies temperature in figure 4 and 5. Line 262, it isassumed that the temperature is linearly varying along the flow direction. But why thisis necessary, how this is justified ? It should be clarified and explained. Some equa-tions should be checked, in particular equation 1b which is incorrect in this form. ResponseïijŽ The assumptions for the linear and radial flows are: In a horizontal or an inclined aquifer,radial thermal flow also has several assumptions for its features. The aquifer is continually distributed with a uniform thickness. The permeability of the aquifer is uniform, leading to inference of an uniform thermal property. The aquifer is assumed to have a single source, leading to the fact that the temperature field with smooth temperature variations.

In Figures 4 and 5 the temperature changes are smooth and gradual, with the aquifer is assumed to have only one heat source. Therefore, the assumption for temperature is not necessary to be linear.

We have checked the formulations in all the equations. We have removed the equation (1b). Equation (1b) was not used actually used any way; it is there for completeness of the flow of none steady state.

CommentïijŽ 5) Itis difficult to understand from section 6 and figure 5 how analytical results from section 5 are applied to the field case. This should be clarified as well as the geometry of the aquifer and the assumptions used for the calculations. ResponseïijŽ We added text to clarify the utility of analytical solution to the field site. We added a cross section of the field site to enhance understanding the of fault zone (Fig.6 in
revised version). The fault plane for the deep fault is assumed to have a flat surface, slightly curved to the south at the western end, based on borehole data in the field site. Assumptions are made for applications of analytical solutions (Eqs. 11,12) to the Xinzhou geothermal field. The fault is approximated to have uniform properties such as thickness, permeability. The temperature distribution between boreholes is resulted from a single heat source so that the temperature changes are smooth and the trend is known. In this case study we used the linear interpolation to approximate the actual inter-borehole temperature. The linear approximation for temperature is expected to have negligible error, because of the steady state nature of the flow field condition. A more rigorous approximation could have obtained a better fit for temperature, for example, a second-order of polynomial function. But it is hard to find the data for the fit of the curve.

We added the above text in the revision to the analytical sections ( (Section 4.1; 4.2, revised version)

CommentïijŽ 6) How is calculated the faultpermeability ? Permeability values are given (line 392) and largely discussed (section7.2). It appears as a major result (line 45-46 in the abstract and 475-477 in the conclu-sion), but it should be explained how permeability is calculated. ResponseïijŽ The fault permeability calculation is discussed in details (Section 5.3, revised version). The assumptions are also provided.

CommentïijŽ 7) Moreover, in suchpermeable fault zone, with such temperature gradi-ents, one may expect natural freeconvection to occur. This should be discussed since conditions for flow and pressuredistribution in the fault zone (slanted aquifer) should control the response observed inthe deep borehole. Interactive comment on Hydrol. Earth Syst. Sci. Discuss., https://doi.org/10.5194/hess-2018-624, 2019 ResponseïijŽ In light of the flow field with the permeable fault, one would think whether the tem-perature and pressure condition could lead natural free convection to occur. The flow condition in the top part of the deep fault system is predominantly upward. There is no field evidence that natural convection occurs in this field site at the scale of this size

scale of about 800 m across.A simulation in regional scale of the deep fault reaching 7 km deep, the flow field shows that the upper part is dominantly up-swelling flow while, the down-drawing flow occurs near the bottom part of the deep fault. More detailed investigation is referred to Lu et al. (2017). We added the above text in the revision to the end of permeability calculation section (Section 6.3 in the revised version).

Lu, G., Wang, X, Xu,F., Li,F., Wang,Y., Qi,S.,Yuen,D.: Deep geothermal processes act through deep fault and solid tide in Xinzhou geothermal field in coastal Guangdong,China, Physics of the Earth and Planetary Interiors, 264, 76–88, 2017, http://dx.doi.org/10.1016/j.pepi.2016.12.004.